# The Autonomic Nervous System Differentiates between Levels of Motor Intent and End Effector

**DOI:** 10.3390/jpm10030076

**Published:** 2020-07-31

**Authors:** Jihye Ryu, Elizabeth Torres

**Affiliations:** 1Psychology Department, Rutgers University Center for Cognitive Science, Rutgers University, Piscataway, NJ 08854, USA; jr1102@psych.rutgers.edu; 2Psychology Department, Rutgers University Center for Cognitive Science, Computational Biomedicine Imaging and Modeling Center at Computer Science Department, Rutgers University, Piscataway, NJ 08854, USA

**Keywords:** embodied cognition, agency, action ownership, network analysis, graph theory, motor control, voluntary motion, precision medicine

## Abstract

While attempting to bridge motor control and cognitive science, the nascent field of embodied cognition has primarily addressed intended, goal-oriented actions. Less explored, however, have been unintended motions. Such movements tend to occur largely beneath awareness, while contributing to the spontaneous control of redundant degrees of freedom across the body in motion. We posit that the consequences of such unintended actions implicitly contribute to our autonomous sense of action ownership and agency. We question whether biorhythmic activities from these motions are separable from those which intentionally occur. Here we find that fluctuations in the biorhythmic activities of the nervous systems can unambiguously differentiate across levels of intent. More important yet, this differentiation is remarkable when we examine the fluctuations in biorhythmic activity from the autonomic nervous systems. We find that when the action is intended, the heart signal leads the body kinematics signals; but when the action segment spontaneously occurs without instructions, the heart signal lags the bodily kinematics signals. We conclude that the autonomic nervous system can differentiate levels of intent. Our results are discussed while considering their potential translational value.

## 1. Introduction

The field of embodied cognition (EC) has provided a powerful theoretical framework amenable to bridge the gap between research probing our mental states and research investigating our physical actions [1,2,3]. Indeed, within the framework of EC, the construct of agency conceived as a cognitive movement phenomenon [4,5,6] may provide a way to finally connect the disparate fields of cognitive science and motor control. An important component of agency is action ownership [6,7,8], i.e., the sense that sensory consequences of the actor’s action are intrinsically part of the actor’s inner sensations. When the actor owns the action, s/he has full control over those sensations that are internally self-generated and self-monitored by the actor’s brain and yet extrinsically modulated by external sensory goals. A critical aspect of this internal–external loop is the identification of the level of actor’s intent and its differential contribution to the action’s intended and unintended sensory consequences.

In recent years, a body of knowledge has increased our understanding on the sensory consequences derived from intentional actions, as such action components deliver an overall sense of agency [9,10] through elements of body-ownership closely interrelated with motor control [11,12]. Less explored, however, have been parts of the action that are unintended or that transpire spontaneously and largely beneath awareness. Such actions’ components exist at the involuntary (uncontrolled, random motions) and at the autonomic (pacemaker, periodic motions) levels of neuromotor control (Figure 1). They do not require explicit instructions or precisely defined external goals, yet they too contribute to the differentiation of levels of intent in our actions [13,14]. More importantly, at the cognitive level of decision making, these unintended movements contribute to the acquisition of decision accuracy within the context of the type of motor learning that is induced by different cognitive loads [15,16].

At the motor control level, autonomous and spontaneous movements are important to develop a sense of action ownership in the face of motor redundancy [17]. Spontaneous motions can be covert, as those subtle motions occurring in a coma patient [18] or those occurring in a neonate [19]; or overt, as when they coexist with deliberate/staged ones, embedded in complex sports routines [13,14] and/or ballet choreographies [20]. Such complex overt movements require the coordination and control of many degrees of freedom (DoFs) across the body. Thus, as we produce fluid and timely goal-oriented actions, kinematic synergies self-emerge and dynamically recruit and release the bodily DoFs, according to task demands [21,22,23]. Conscious decisions generating movements that attain external goals take place as the brain interweaves deliberate and spontaneous movement segments. Such segments in our complex actions gracefully build an ebb and flow of actions intended to a goal, and sensory consequences from those actions [13]. Some of these sensations that voluntary movements give rise to [24] return to the brain as feedback from the intentional part of the movements, thought to contribute to our internal models of action dynamics [25,26]. This form of volitionally controlled kinesthetic reafference cumulatively helps us build accurate predictions of those intended sensory consequences [24], while other unintended movements return to the brain as spontaneous reafference in the precise sense that they do not follow from instructed acts. These spontaneous activities provide contextual cues that support motor learning, motor adaptation and action generalization across different situations [13], including pathologies of the nervous systems [27,28].

One informative aspect of this ebb and flow of intent and spontaneity in our actions is the fundamental differences that emerge in the geometric features of the positional trajectories that the moving body describes [29,30,31]. When the motions are intended, geometric measures related to path curvature and path length show invariance to changes in movement dynamics, i.e., these metrics remain robust to changes in speed, mass, etc. [21,31,32,33,34]. In contrast, trajectories from unintended motions produce different signatures of motor variability bound to return to the brain as *spontaneous feedback*. These internal sensations could help the brain differentiate contextual variations emerging from external environmental cues from sensory information that is internally self-generated by the nervous systems [13,14]. External information may include, for example, changes in visual and auditory inputs, such as shifts in lighting conditions, or modulations in sound and music [20,35]. 

The geometry of the uninstructed spontaneous movements’ trajectories dramatically changes with fluctuations in the movements’ dynamics. Changes in speed [21,30,31,32] or mass [14] affect their motor variability in fundamentally different ways (if we compare the signatures of variability derived from the spontaneous samples to those derived from deliberately staging the same movement trajectories [14,32].) More importantly, the fluctuations in the motor variability of these spontaneous motions can forecast symptoms of Parkinson’s disease before the onset of high severity [27,36]. They have also aided in evoking the sense of action ownership and agency in young pre-verbal children [28]. For these reasons, here we posit that deliberate and spontaneous segments of complex covert actions ought to differentially contribute to our physical sense of action ownership and to our overall sense of agency. To examine this proposition, we follow a phylogenetically orderly taxonomy of the nervous systems’ maturation (Figure 1B) and examine all levels of neuromotor control—from autonomic to deliberate—necessary to coordinate voluntary motions (Figure 1A).

More specifically, since autonomic systems are vital to our survival and wellbeing, they may remain impervious to subtle distinctions between deliberate and spontaneous motions that take place across the body, as the end effector completes goal-directed actions. Here we explore the interplay between autonomic signals and voluntary motor control in actions that integrate deliberate and spontaneous motions across the body. We use a new unifying statistical framework for individualized behavioral analyses and network connectivity analyses and offer a quantitative account of how these movement classes contribute to the overall embodied sense of agency.

## 2. Materials and Methods

### 2.1. Experimental Design

#### 2.1.1. Participants

Nine undergraduate students (2 males and 7 females) between the ages of 18 and 22 years were recruited from the Rutgers human subject pool system. Two were left-handed and seven were right-handed, and all had normal or corrected-to-normal vision. All participants received credit for their participation, and provided informed consent, which was approved by the Rutgers University Institutional Review Board. The study took place at the Sensory Motor Integration Lab at Rutgers University.

During the experiment, movement kinematics and heart signals were recorded from each participant. However, one participant’s recording had too much noise (i.e., inaccurate sensor position with error larger than 10 cm), so we excluded this participant’s data in the analysis. For that reason, eight participants’ motor and heart signals were analyzed.

#### 2.1.2. Sensor Devices

Motion capture system (kinematics data): Fifteen electromagnetic sensors sampling at a frequency of 240 Hz (Polhemus Liberty, Colchester, VT, USA) were attached to the participant’s upper body in the following locations: center of the forehead, thoracic vertebrate T7, right and left scapula, right and left upper arm, right and left forearm, performing hand, and the performing hand’s index finger. These sensors were secured with sports bands to allow unrestricted movement during the recordings. Motor signals were recorded in real-time by Motion Monitor (Innovative Sports Training Inc., Chicago, IL, USA) software, where the participant’s body was constructed by a biomechanical model, and movement data were preprocessed by an embedded filtering algorithm of the software, providing the position and kinematics of each sensor.

Electrocardiogram (heart data): Three sensors of electrocardiogram (ECG) from a wireless Nexus-10 device (Mind Media BV, Herten, The Netherlands) and Nexus 10 software Biotrace (Version 2015B) were used to record heart activity. At a sampling rate of 256 Hz, the sensors were placed across the chest according to a standardized lead II method.

#### 2.1.3. Experimental Procedure

Participants sat at a desk facing an iPad tablet (Apple, Cupertino, CA, USA), which was used to display stimuli during the experiment, and participants responded by touching the tablet screen. The tablet display was controlled with an in-house developed MATLAB (Release 2015b, The MathWorks, Inc., Natick, MA, USA) program and TeamViewer application (Germany).

As shown in Figure 2, for each trial, the participant was presented with a circle on the tablet screen. This circle served as a prompt for the participant to touch the tablet screen within five seconds. After the touch, either 100 ms, 400 ms, or 700 ms elapsed, and the participant heard a tone at 1000 Hz for 100 ms. Then, on the tablet screen, the participant was presented with a sliding scale, ranging from 0 to 1 (second), to indicate how long he/she perceived the time elapsed between the touch and the tone. The response was to be made within five seconds upon the display of the sliding scale. The five second time-window was considered enough for the participant to provide a response, as it took approximately 1 s to touch the screen and retract the hand back to its original position. There was a total of three conditions, namely control, low cognitive load, and high cognitive load, and each condition consisted of 60 trials. In the control condition, the participant simply performed each trial with no additional task; under the low cognitive load condition, the participant performed each trial while repeatedly counting forward 1 through 5; under the high cognitive load condition, they counted backwards from 400 subtracting by 3 while they performed each trial. Participants counted forward and backward at their own comfortable pace, and they took breaks in between each condition. The experiment set up took about 30 min, and the recording took about 40 min.

### 2.2. Statistical Analysis Overview

#### 2.2.1. Preprocessing

In this study, we extracted the kinematics (i.e., linear speed, angular acceleration) and heart data during time segments when the participant made a pointing motion towards the circle presented on the tablet screen, and we combined them across the three conditions. As a result, we analyzed the kinematics and heart data recorded while the participant made 180 pointing motions (less any trials that were deemed noisy; the most trials we excluded per participant due to instrumentation noise were 12 trials).

To analyze the ECG and kinematics data in tandem, we up-sampled the kinematics data from 240 Hz to 256 Hz using piecewise cubic spline interpolation. Note, the ECG signals were not synchronized with the kinematics data but were manually time stamped at the start and end of each experimental condition. For that matter, we expected the presence of a lag between the two modes of signals—kinematics and ECG—but the lag did not exceed 1 s.

To exclude effects of muscle motion from the ECG heart data, we bandpass filtered the data with Butterworth IIR for 5–30Hz at 2nd order. This filter was effective in identifying QRS complexes and extracting R-peaks in previous studies [16,37]. Here, the filter excluded the dominant frequency range where typical kinematics signals are present (see Appendix A
Figure A1). We performed our analyses using both filtered and non-filtered EKG data and found similar trends and patterns. However, the paper only presents the results from using the filtered data, as it is a better reflection of the heart activity.

#### 2.2.2. Data Analysis Structure

We used the rationale in Figure 1 to structure our analyses, with a focus of two main axes denoting the level of motor intent and awareness that the brain may have during complex tasks (Figure 3A). More precisely, one axis explored possible differentiations between time segments of the pointing movements that were deliberately aimed at an external target (forward/high motor intent) vs. segments that were consequential to the deliberate ones (backward/low motor intent). The latter may occur when the hand retracts back to rest, or when after touching the target the person transitions the hand in route to another goal-directed motion. These segments have been studied in our lab across very complex motions in sports (boxing, tennis) and in the performing arts (ballet, salsa dancing). We have coined them spontaneous movements and discovered that they have precise signatures that distinguish them from the deliberate ones. For this reason, we hypothesized here that these spontaneous motions would have different stochastic signatures or be differentially expressed in relation to the deliberate ones.

The other axis explored possible contributions of body parts that were not directly related to the end effector (the performing hand) executing the pointing task. We reasoned that there may be higher motor intent devoted to the performing hand of the participant than to the non-performing side of the body. Furthermore, we explored how other body parts (also co-registered within the sensors’ network) contributed to the overall performance of this task.

These two axes were explored at the voluntary level of motor control interleaving deliberate goal-directed (forward) actions and spontaneous (backward) segments of the full pointing loop. We also included in our analyses the autonomic level of control in the taxonomy of Figure 1A, and to that end, we co-registered the heart activity and incorporated it into the bodily kinematics activity (Figure 3B). We next explain how to overcome challenges in sensor data fusion from disparate systems along with new approaches to analyze these multi-modal data.

#### 2.2.3. Challenges of Multilayered Data with Non-Linear Dynamics and Non-Normally Distributed Parameters

Disparate physical units: Different instruments to assess biorhythms from different layers of the nervous system (i.e., kinematics vs. EKG) output biosignals with different physical units (e.g., m/s from the kinematics speed, mV from the EKG). This poses a challenge to integrate these signals and examine their interrelations across these layers.

Allometric effects: Another issue is that when examining such data from different participants with different anatomical sizes, allometric effects may confound our results. This is so because, e.g., the speed ranges that a person attains depend on the length of the arm. Longer arms tend to broaden the ranges of speed and contribute to the distribution of speed values that the person attains in any given experiment. As such, we needed to account for these possible allometric effects.

Assumption of normality: Another related matter to the ranges of speed and their distributions is that they vary from person to person according to multiple factors (e.g., age, body mass, sex, fitness, etc.) [38]. These variations result in probability distributions with heavy tails, which are incompatible with common assumptions of normality in the literature. When the effects of the task, or the inherent motor noise in the system, are such that most values related to the speed distribute more densely toward the left of the frequency histogram (e.g., in autism exponentially distributed maximum speed amplitude is common [39]), assuming normality may incur spurious results. This is so because speed ranges from 0 to some limiting value for each person (the maximum speed that the person can reach before damaging the joints). As such, when one obtains the mean ± two standard deviation values to approximate standard error bars (which is very common in the motor control literature) while summarizing the statistical features of the data, the data may fall in the negative speed ranges (which is physically absurd).

Assessing similarity in probability space: Going beyond significant hypothesis testing models, one may need to assess the differences between probability distributions. To that end, one may need a proper similarity metric. Yet, when our data represent points in probability space, and the distributions are not symmetric, it is challenging to assess their similarity in a consistently proper way. Measures like the Fisher information metric are designed to compare symmetric distributions and the Kullback–Leibler divergence is computed asymmetrically between distributions (one-sided). We would like to have a proper (two-sided) distance metric to assess change and its rate when points are related to non-symmetric continuous probability density functions, or to their discrete approximations.

Degrees of freedom across intent levels of motor control: Multiple locations of the grid of sensors, co-registering biorhythms from different nervous systems, contribute differently to the overall behavior of the system. Some may be more directly related to action success, while others may provide support. Separating the bodily region within a kinematics-heart network can be challenging because of the non-linear dynamics of the interactive systems. Yet, most methods assume or impose local linearities to model such phenomena. Here we propose to approach this problem by treating the grid of sensors as a dynamically evolving weighted interconnected network, whereby we track self-emerging modules informing us of spontaneous synergies and connectivity patterns.

#### 2.2.4. Some Solutions to the Challenges

New data type for disparate physical units: We created a data type called the micro-movement spikes (MMS), which is a unitless, standardized waveform derived from the moment to moment fluctuations in the raw data peaks’ amplitude and/or timing. This data type extracts the fluctuations in amplitude and/or timing of any waveform with peaks and valleys (e.g., time series of speed values or kinematic related values derived from them). To that end, we obtained the empirically estimated moments from the peaks in the raw waveform. We then built a new waveform that can be normalized according to various criteria. This new waveform is then unitless and refers to a relative quantity (rather than to an absolute quantity).

Data standardization to account for allometric effects: The anthropology and paleontology literature has several solutions to address comparative data that may come from different bone sizes across, e.g., different humanoids [40,41]. Equation (1) provides an example of standardization to scale values derived from any waveform with peaks and valleys, which can be derived, e.g., from data series with different physical units, from effectors of different sizes.
(1)$$StandardizedPeak=\frac{LocalPeak}{LocalPeak+Avrg_{min-to-min}}$$

The standardized quantities are in the real-valued [0,1] interval. They are coined MMS amplitudes and treated as a continuous random process. We characterized several complex behaviors from various layers of the nervous systems using the MMS and expressed them in two forms: (1) without preserving the original frames of the data, i.e., just focusing on the MMS amplitude fluctuations, and (2) conserving the original frames, in which case, we would 0-pad those that are not spikes or preserve their values as additional gross data contributing to the phenomena in question. Either way, these fluctuations ought not to be averaged out by assumptions of normality. Whereas in the extant literature these fluctuations are considered noise or superfluous, here we treated them as important signals.

Distribution-free approach to counter current assumption of normality: We did not assume normality in the data. Instead, we gathered enough data to empirically estimate the best family of probability distributions that fits the data. To that end, we here used maximum likelihood estimation (MLE) with 95% confidence intervals and sought the best continuous family that fit our data.

Distance metric to assess similarity in probability space: We here introduced the use of the earth mover’s distance metric (EMD) [42,43,44] to approximate (using the frequency histograms of the MMS amplitudes) the stochastic shifts in probability space that occur for different movement types. This is an appropriate similarity metric that allowed us to examine the extent to which different levels of motor control change the stochastic patterns. We briefly describe it below:

The EMD, also known as the Kantarovich–Wasserstein distance [45], measures the distance between two discrete probability distributions. Given two discrete distributions P = {(p_1_,w_p1_), … (p_m_,w_pm_)}, where pi is the cluster representative and w_pi_ is the weight of the cluster; and Q = {(p_1_,w_p1_), … (p_n_,w_pn_)}, EMD computes how much mass is needed to transform one distribution into another. Defining D [d_ij_] as the ground distance matrix, where d_ij_ is the ground distance between clusters p_i_ and q_j_, and F = [f_ij_] with f_ij_ as the flow between p_i_ and q_j_; EMD is computed by minimizing the overall cost of such:$$\mathrm{Work}\;(P,Z,F)=\sum_{i=1}^{m}\sum_{j=1}^{n}d_{ij}f_{ij}$$

As there are infinite ways to do this, the following constraints are imposed to yield EMD values:$$f_{ij}\geq 0\;1\leq i\leq m,1\leq j\leq n$$
$$\overset{n}{\underset{j=1}{\sum }}f_{ij}\leq w_{p_{i}}\;1\leq i\leq m$$
$$\overset{m}{\underset{j=1}{\sum }}f_{ij}\leq w_{q_{i}}\;1\leq j\leq n$$
$$\sum_{i=1}^{m}\sum_{j=1}^{n}f_{ij}=\min(\sum_{i=1}^{m}w_{p_{i}},\sum_{j=1}^{n}w_{q_{j}})$$
$$\mathrm{EMD}(P,Q)=\frac{\sum_{i=1}^{m}\sum_{j=1}^{n}d_{ij}f_{ij}}{\sum_{i=1}^{m}\sum_{j=1}^{n}f_{ij}}$$

Network connectivity analyses to assess degrees of freedom recruitment across modalities of motor control: We used graph theory to examine the inter-relations across the nodes of the multilayered kinematics-heart network. To that end, we derived an adjacency metric of pairwise quantities reflecting the cross-correlation between any pair of nodes in the grid. We then constructed weighted directed networks and borrowed connectivity metrics from brain-related research. We extended these methods to represent the peripheral network using the bodily biorhythms from multiple layers of the nervous systems’ functioning, spanning from voluntary to autonomic (Figure 1A).

#### 2.2.5. Choice of Kinematics Parameter

The recording of positions over time across 10 upper body parts allowed us to estimate two aspects of the biorhythmic data: spatial and temporal aspects, both of which are critical to characterize proper coordination and control. A parameter encompassing both aspects is the velocity. The derivative of position over time creates vector fields with direction and extent. Each point in the field (along the velocity trajectory) occurs in time and moves in space.

To assess spatial components, we used the scalar speed (distance traveled per unit time, where the unit time is taken constantly at the rate of 240 frames per second). We used the Euclidean norm to compute the length of the velocity vector at each unit time, thus quantifying the rate of change in position per unit time—the linear speed (m/s). Likewise, we used the orientation data from each sensor and obtained the angular velocity from the rotations of each body part. Using appropriately the quaternion representation of rotations and the Euclidean metric to quantify the magnitude of the angular velocity vector, we obtained the angular speed (deg/s). These waveforms derived from the first order change are useful, but at the time scale (~1/2 h) of our experimental assay, they provided fewer peaks per trial than waveforms derived from the second order change (i.e., linear acceleration (m/s^2^) or angular acceleration (deg/s^2^)).

As we needed many spikes for our distribution-fitting and stochastic analyses, we used the angular acceleration kinematics data. Note, it was possible to have had participants perform more trials to obtain a larger number of spikes using the linear speed; however, this would have fatigued the participants as the length of the experiment was around 70 min (inclusive of 40 min for set up). For that reason, within this amount of time, it was ideal to use the angular acceleration as our kinematic parameter of interest. This choice of parameter to analyze the stochastic patterns of the moment by moment fluctuations in signal amplitude (i.e., the spatial component of our analysis) provided a tighter confidence interval in the empirical estimation of the best probability distribution family fitting the data.

We also examined temporal components of the data. To that end, we used the linear speed patterns and the cross-correlation function. We extended our analyses to different kinematics parameters, and while they all showed similar patterns and trends, we found the linear speed to best characterize the differing patterns of motor intent. For that reason, we presented the results of the temporal analyses involving cross-correlation based network connectivity patterns using the linear speed as our waveform of choice (Figure 3C).

### 2.3. Data Analysis on Kinematics Network Connectivity

As a first step, we separated the kinematics data obtained from all 10 body parts, using the start and end time of the performing hand making a forward–deliberate motion, and the hand making a backward–spontaneous motion (Figure 4A). This was possible to do (automatically) because (1) the speed was near 0 at the onset of the motion towards the target; (2) the distance to the target monotonically decreased and once again the hand paused at the target at near 0 speed. As the deliberate (forward) segment was completed, the speed rose again away from 0, and the distance to the target increased as the hand followed the backward segment of the full pointing loop. The two segments could be automatically differentiated also because the deliberate (forward) one was less variable than the spontaneous (backward) one [14].

For the connectivity analysis centered on spatial aspects of the signal amplitude, we pooled the angular acceleration data from each body part and extracted the MMS amplitudes (referred to as MMS from here on). We then built frequency histograms of the MMS and explored several families of PDFs using MLE. The continuous family of Gamma PDFs yielded the best fit (Figure A2) and served to provide the noise-to-signal ratio (NSR; computed to equal the Gamma scale parameter) for each body part (Figure 4B,D–F). These were then visualized as node size in the schematics of the network in Figure 4K across different motor intent levels.

To characterize the connectivity of 2 body parts, we took the pairwise absolute difference between angular acceleration and based on the obtained absolute difference time series, computed the corresponding MMS. We then fitted the Gamma scale parameters (i.e., NSR) (Figure 4C–F), which were visualized as edges in the schematics of the network in Figure 4K. The intuition behind taking the absolute difference in angular acceleration time series from two body parts is that this reflects the change in positional distance between those two body parts and thus represents the connectivity (physical distance) between those two. The NSR values were then compared between different movement segments (i.e., forward vs. backward) and different sides (i.e., performing vs. non-performing arm/hand), to understand the noise level during different levels of motor intent. Note, for each type of motor segment (i.e., forward vs. backward), and for each side (i.e., performing vs. non-performing), more than 2500 spike amplitude data were extracted. These spike amplitude data were then plotted on a frequency histogram using Freedman–Diaconis binning rule [46]. They were used for empirical estimation of the best PDF in an MLE sense. The results yielded the Gamma probability distribution function (PDF) (see Figure A2B).

Connectivity analyses on temporal aspects of coordination involved the linear speed from each pair of body parts. We computed pairwise cross-correlations to derive an adjacency matrix that would represent a weighted undirected graph. Here, the ij-link’s weight is the maximum cross-correlation value between nodes i and j (that is, the corresponding two body parts). From these matrices, we computed clustering coefficients, which are measures that characterize the local connectivity (i.e., functional segregation). They would represent self-emerging kinematic synergies. Specifically, the degree of a node in the network (number of links at a node) between a set of nodes form triangles, and the fraction of triangle numbers formed around each node is known as the clustering coefficient (Figure 4G–J). This measure essentially reflects the proportion of the node’s neighbors (i.e., nodes that are one degree away from the node of interest) that are also neighbors of each other [47]. Here, we computed the average intensity (geometric mean) of all triangles associated with each node, where the triangles reflect the degree strength, and is computed as shown below (using an algorithm by [48]; Equation (2)).
(2)$$C_{i}=\underset{i\in N}{\sum }\frac{t_{i}}{k_{i}\left(k_{i}-1\right)}$$


N: set of all nodes (composed of 10 body parts)$C_{i}$: cluster coefficient for node i (i∈N)$t_{i}$: geometric mean of triangles links formed around node i (i∈N)$k_{i}$: number of degrees (links) formed around node i (i∈N)


To visualize the network, we represented the median pair-wise cross-correlation values as the edge thickness and median cluster coefficient values as the node size (Figure 4K). The median cross-correlation and cluster coefficient values were then compared between different movement segments (i.e., forward vs. backward) and different sides (i.e., performing vs. non-performing arm/hand) to understand how linear correlations differed across varying levels of motor control.

### 2.4. Data Analysis on Kinematics-Heart Network Connectivity

As with the kinematics connectivity analysis, we segmented the data of the filtered EKG data along with the kinematics data by the time intervals when the performing hand was making a deliberate forward motion and a spontaneous backward motion (Figure 5A).

For the spatial domain of connectivity, we took the segmented data of angular acceleration and EKG data, extracted MMS from both signals, and plotted a histogram of the MMS. Because the MMS of EKG signals did not follow a Gamma distribution, in order to assess the connectivity between the two, we computed the earth mover’s distance (EMD) between the histogram from a single body part and from the EKG data (Figure 5A–D).

For the temporal domain, we computed pairwise cross-correlations, along with lag, between the EKG filtered time series and each body part’s linear velocity time series. In fact, in our analysis, we found an interesting pattern in directionality (i.e., lag) of correlation and deemed it informative to present them in the network graph. For that reason, edge thickness was represented by the median cross-correlation values, and color of the edges were visualized, where red indicated EKG signals leading linear velocity signals, and blue indicated linear velocity leading EKG signals (Figure 5G).

For all these metrics, we compared the medians between different movement segments (i.e., forward vs. backward) and different sides (i.e., performing vs. non-performing arm/hand), to understand how stochasticity and temporal dynamics changed across varying levels of motor intent between the heart (from ANS) and kinematics (from PNS/CNS).

## 3. Results

### 3.1. Higher Motor Intent Results in Higher NSR in Spatial Parameters

Motor intent in the context of our experimental assay specifically refers to the level of deliberateness (or spontaneity) of the movement segment in route to an external target (away from it). An instructed pointing action to touch the target is a goal-directed reach with high level of intent. In contrast, the uninstructed spontaneous retraction away from the target carries lower motor intent than the goal-directed one.

As a first set of analysis, the MMS extracted from the angular acceleration data from each body part were aggregated across all trials and conditions and arranged by different movement segments (forward–deliberate vs. backward–spontaneous) and different sides (performing vs. non-performing). The same was also done on the MMS extracted from the absolute difference in angular acceleration from all pairs of body parts. The NSR was found to be significantly higher when the motions were deliberate and on the performing side (Figure 6).

Specifically, NSRs of the kinematics time series from each body part shown was highest when an individual exerted higher motor control under higher level of motor intent, such as on the performing side of the body and during a forward–deliberate motion. Conversely, when an individual did not deliberately intend to move the arm, as exhibited on the non-performing side of the body and during a backward–spontaneous motion, the NSR was at its lowest. The NSRs for all pairs of body parts’ absolute difference in angular acceleration (i.e., change in distance between the pairs of body parts), on the other hand, was higher on the performing side (vs. non-performing side) but did not show such a consistent pattern when comparing between the two motion segments (forward vs. backward). Details of the 95% confidence interval of the fitted Gamma scale parameter (i.e., the NSR) for all participants, all body parts (Figure A3), and all pairs of body parts (Figure A4) can be found in the Appendix A.

### 3.2. Higher Motor Intent Results in Higher Cross-Correlations and Clustering of Temporal Parameters

We used the MATLAB Network Connectivity toolbox [49] and examined the adjacency matrix derived from the pairwise maximal cross-correlation coefficient based on the time series of linear speed values. The clustering coefficient (CC) was obtained for each body part as a metric of functional segregation. For analysis, we examined the median cross-correlation values as a function of the CC values. Here we found that higher level of motor intent (i.e., during forward–deliberate motion performed with the performing hand) resulted in a tendency of increased CC and increased median cross-correlation values (Figure 7).

When we compared between different motion segments, median cross-correlations were higher for forward motions than for backward ones for all but two participants. When we compared between different sides, all participants showed higher correlation on the performing side than the non-performing one. The median CC was higher for forward motions than for backward segments for all participants and higher for the performing side than the non-performing side for all but two participants. For all participants, both measures showed statistical significance in their difference (see Table A1 of Appendix A for detailed statistical results).

The distinctions that we observed from these findings, on how different levels of motor intent had separable network connectivity patterns based on temporal aspects of the kinematics data, are consistent with the patterns uncovered using spatial aspects of the kinematics data. Specifically, when we exerted higher intent on our body, regardless of the physical trajectory of the motion, there was a stronger connectivity across our body parts. However, we noted that this pattern was not as uniform across all participants as we had found in the spatial aspect of the network analysis.

### 3.3. Kinematics and EKG (Heart) Signals Show Larger Stochastic Differences for Higher Motor Intent and Control

To assess patterns of connectivity between biophysical signals derived from voluntary and autonomic levels of motor control we examined the kinematics (generated by the CNS–PNS) and the heart activity (generated by the ANS). The patterns of MMS stochasticity and temporal correlation across these systems distinguished levels of motor intent and control.

The analyses involving EKG and kinematics revealed larger stochastic differences in MMS data when higher motor intent and control were exerted. More precisely, the pairwise EMD showed higher differentiation between these two signals in all but one participant when forward motion was made, but only on the performing side of the body. Furthermore, all but two participants showed higher EMD on the performing side of the body, but only during forward motions. On the other hand, however, when backward motion was made, we found an opposite pattern, where all participants showed higher EMD on the non-performing side. We inferred that there may have been a modulating factor that underlied the stochastic relation between kinematics and heart signals.

When we examined the temporal relations between the two signals, by computing pair-wise cross-correlations, we saw higher cross-correlations when there was lower motor intent across all participants, that is, during backward motions and on the non-performing side. Here, we noted the low range of the correlation coefficient values of around 0.1. However, we saw a similar trend when this was based on the non-filtered raw EKG data, with a higher range around 0.6.

### 3.4. EKG Leads Kinematics under Higher Motor Intent, but Opposite Pattern Emerges in Spontaneous Motions Requiring Less Motor Intent

We also examined the lag values to assess which signal leads the other. We found that with motions under higher motor intent (i.e., during forward–deliberate motions performed with the performing side of the arm), EKG signals tended to lead the kinematics signal. On the other hand, in movements performed under lower intent (i.e., during backward–spontaneous motion, and on the non-performing side of the arm), kinematics signals tended to lead the EKG signals. This is depicted in Figure 8.

We caveat that because the EKG device and motion capture system were not exactly synchronized, the absolute lag value may not be as meaningful. Nevertheless, as we analyzed these data in terms of the difference (i.e., the delta lag values between forward and backward motions, and between performing and non-performing sides), it was indeed meaningful to find such patterns uniformly across all participants.

Table 1 summarizes the results that we showed in the sections above. We emphasize that although we examined a small number (eight) of participants, each individual’s data were composed of a significant amount of data points with unique non-Gaussian stochastic characteristics. For that reason, instead of presenting the results with NHST (null hypothesis significant tests), we presented the results by comparing the median difference between data points, from different levels of intent, for each individual.

## 4. Discussion

This paper examined elements of the construct of agency from the embodied cognition framework and dissected several layers of neuromotor control contributing to the sense of action ownership. These layers, defined along a phylogenetically orderly taxonomy of maturation, follow a higher-to-lower gradient of intent, from voluntary, to involuntary, to autonomic signals. At the voluntary level, we followed the instructed–deliberate and the uninstructed–spontaneous segments of the target-directed pointing act, positing that they could differentiate between levels of intent and as such, delineate (from the fluctuations in their biorhythmic activity) when a given movement segment was deliberately performed with intent vs. when the segment happened spontaneously without instruction. This differentiation is important to distinguish the sensory consequences of voluntary acts from those of acts that are not intended or that occur autonomically. The sensory consequences of the latter have not been currently studied, yet they seem important to complement von Holst’s and Mittelstaedt’s principle of reafference as we know it today [24].

Our initial thought was that autonomic systems contributing to our brain’s autonomy over the body and to our overall embodied sense of agency would remain impervious to stochastic shifts at the voluntary levels. We reasoned that given the vital role of these systems for survival, their robust signal would not reflect subtle changes in levels of intent, motor awareness, and voluntary control. As such, our guess was that if during voluntary movements, there were stochastic differences between instructed–deliberate and uninstructed–spontaneous segments of the reach, or between performing and non-performing sides of the body, such shifts in patterns of variability would not be appreciable in the heart signals’ fluctuations. Our guess was altogether wrong. Not only were the heart signal differences quantifiable at the level of micro fluctuations in signal amplitude, these differences were appreciable as well in the inter-dynamics of the kinematics and cardiac signals.

### 4.1. The Autonomic Nervous System Differentiates across Levels of Motor Intent: Implications for Computational Models and Basic Cognitive Neuroscience

We found that when movements are intended and deliberately performed to attain the goal defined by an external (visual) target, the heart signal leads the movement kinematics signal. Yet, when these overt movements are spontaneous in nature, i.e., uninstructed and not pursuing the completion of a specific externally defined task goal, the heart signal lags the movement kinematics signal. Across spatial and temporal parameters, we found consistent trends and confirmed the trends through different parameters. Indeed, deliberate motions, executed with the performing effector, carry higher levels of NSR, denoting higher fluctuations away from the empirically estimated mean.

We interpret these findings considering the principle of reafference, treating micro-movements as a form of re-entrant sensory feedback [24]. Furthermore, we discuss the possible contributions of these self-generated signals to the self-emergence of cognitive agency from motor agency, namely the sense that one can physically realize what one mentally intends to do, confirm the consequences (both intended and unintended), and, as such, mentally own the physical action.

Von Holst and Mittelstaedt studied the complexities of reafference across the nervous systems in the 1950s. They tried to capture the inherent recursiveness that relates movements and their sensations as they flow within closed feedback loops between the external and the internal environments of the organism. They wrote, “Voluntary movements show themselves to be dependent on the returning stream of afference which they themselves cause.” Undeniably, feedback from voluntary movements currently play an important role in theoretical motor control, particularly within the framework of internal models for action [25,26] and more recent models of stochastic feedback control [50]. Central to all these conceptualizations of the motor control problem has been the notion of anticipating the sensory consequences of impending intended actions. Nevertheless, nothing has been said about the consequences of action segments that bear a lower level of intent, which occur spontaneously, or that are altogether occurring autonomously.

The implications of our results are manifold: Modelers and experimenters in motor control do not seem to be aware of self-emergent, uninstructed, spontaneous motions. These motions are rather assumed to be far removed from cognitive processes, perhaps because they transpire largely beneath awareness (although see [13,51] more recently). Yet, unintended consequences from the uninstructed–spontaneous segments of the voluntary action seem as important as those sensory consequences that result from the instructed–deliberate segments. They may serve to inform learning new tasks, adapting to new environmental conditions or situations, and more generally, they may play a role as a surprise factor to aid propel curiosity and/or to stimulate creative, exploratory thinking. They may help make our “invisible” automatic movements visible to the conscious brain planning and controlling them, and/or to the external observer tracking our behaviors.

Neither these models, nor Von Holst’s work considered the contributions of unintended consequences from spontaneous acts quantifiable at different anatomical and physiological layers of the nervous systems, while trying to model the basic problem that the organism faces, i.e., the paradox of understanding the “self”, which entails parsing out external from internal reafference. Without a unifying framework to quantify these multilayered interactions and their contributions to the emergence of the notion of self, it becomes rather challenging to bridge the cognitive sense of agency, and more basically of action ownership, “I can do this!; It’s me who’s doing this!”, with the type of autonomous motor control that enables successful self-initiation and completion of the intended act. We argue that inclusion of the unintended consequences from overt spontaneous motions and autonomic signals in our models of motor control will help define embodied agency and provide a new framework to objectively quantify it.

The present work provides empirical evidence that (1) different levels of cognitive intent, awareness, and control are indeed embodied and quantifiable in natural, unconstrained movements, and (2) there are important contributions to central cognitive control quantifiable at the periphery in spontaneous segments of our motions *and their consequences*, but also in motions from supporting (non-performing) body parts. Importantly, such differentiating contributions are also present in patterns from signals generated by the autonomic nervous systems. Recent work from our lab has examined cognitive load in relation to autonomic signals and found systematic changes bound to impact the type of feedback that these signals mediated by cardiac muscles generate within the nervous systems [16,52]. These aspects of the motor control problem are not considered at present in any of the mathematical and computational frameworks used to model the human brain, despite a body of empirical data differentiating classes of movements that are less sensitive to changes in dynamics [14,21,30,31,32,33,34] from those which are dynamic dependent [14].

Our work augments Von Holst’s and Mittelstaedt’s principle of reafference nontrivially by including reafferent contributions from other layers of the nervous systems (Figure 1A) and highlighting the need to update our conceptualization of internal models for action. In the past, the literature has focused on voluntary control and goal-directed behavior to define and to characterize agency [5,6,8,11]. However, if new generations of AI models aim to attain artificial autonomous agents with real agency, it may be necessary to reformulate our models and reconceptualize our experiments in embodied cognition to encompass these multiple layers of intent, awareness, and motor function. These results provide a way to distinguish levels of intent in the stochastic feedback from a robust (autonomic signal), as an important addition to prior work distinguishing levels in more variable speed and acceleration signals [14].

### 4.2. Distinguishing Performing vs. Non-Performing End Effector

Another aspect of this work explored the differentiation between the performing end effector and the non-performing one, within the context of connectivity network analyses and levels of NSR. There we found that the micro fluctuations in kinematics activity taken as a weighted directed graph representing an interconnected network of nodes (body parts), can automatically reveal which side of the body is performing the goal-directed task with intent vs. which side of the body is performing the uninstructed spontaneous segment. The importance of this result is several-fold: First, it demonstrates that we can gain information by considering arm movements within a broader context of bodily motions, treated as a fully interconnected network, rather than examining the end effector in isolation. The network connectivity analyses presented here adapt and extend similar methods used in brain analyses to full bodily motions. This is important to connect data from the CNS and the PNS within a full network (see here [52]) and infer the contributions of different bodily sides on the planning, execution, and coordination of the many DoFs of the body in motion. We need these empirical data to improve our multi-layered generative analytical models of neuromotor control involving different spaces of joints and end effectors [21,29,32,33]. Secondly, these results underscore the importance of not eliminating gross data through grand averaging methods that assume a priori a probability distribution and take theoretical means across all data. Here we personalize the analyses and for each participant, we examine the micro fluctuations (away from empirically estimated distribution moments) contributed by both the performing and the non-performing limbs. We do so within the context of full body macro- and micro-motions, thus considering the value of the NSR derived from the gross data that is often thrown away as superfluous noise. The importance of these new methods is that we can examine possible asymmetries in neurological disorders like Parkinson’s disease, where, e.g., tremor may emerge at the performing hand and yet be forecasted in the NSR of activity recorded in the non-performing limbs. Differentiating performing from non-performing NSR in the context of voluntary motions is now possible using these new statistical methods and network connectivity analyses adapted to full body motions. This type of data is rarely examined in clinical work. Lastly, we offer new ways to examine kinematics synergies and possible patterns of co-articulation across the body, while examining the outcomes from the traditional pointing task now extended to also include the spontaneously retracting segments of the full pointing loop.

### 4.3. Implications of the Results for Translational Cognitive Science

An area where these results could be relevant is smart health and AI, connecting digital biomarkers with clinical observational criteria (e.g., [53]). In the clinical world, there are many problems that will require us to be mindful of this intended vs. unintended dichotomy, as there are phenomena that occur spontaneously and largely beneath awareness. It is difficult to model these phenomena within the voluntary reafference framework. The type of reafference that we need to model those problems belongs in the realm of self-emerging aspects of naturalistic behaviors. Among those which are disrupted due to pathologies of the nervous systems are sudden freezing of gait in Parkinson’s disease, leading to the loss of balance and occasional falls; seizures across a broad range of disorders; heart attacks; a subset of repetitive behaviors and self-injurious or aggressive episodes in autism; among others. All these episodes have in common the element of surprise connected to their spontaneity. Several new emerging areas in basic research with a focus on the relationships between property and agency in neurological disorders can also be incorporated in new AI concepts for smart health [15,19,28,53,54,55].

No algorithm relying exclusively on intentional control signals can appropriately capture the essence of these phenomena. To properly characterize it, forecast it, and quickly detect it, we need veridical generative models that understand the differences between the consequences of something that was intended and under voluntary control, something that spontaneously happened, and something that happens autonomically, with high accuracy. We do not have autonomous robots with embodied agency yet, because their staged motions are mostly pre-programmed. These programs may only mimic the predictive consequences of voluntary actions. Self-correcting robotic systems, where such behaviors spontaneously self-emerge, are less common. It is perhaps self-emerging awareness derived from the consequences of spontaneous and autonomic phenomena that makes our embodied agency a special human trait contributing to intelligent control. This type of control, combining deliberate and spontaneous acts, may produce solutions that are capable of generalizing from a small set of specific situations; transfer the learning from one context to another (using contextual variations); and retain robustness to potential interference from new situations in unknown contexts. In future research, it will be important to understand how the type of differentiation that we discovered here, paired with externally vs. internally generated rewards, may contribute to the fast or slow acquisition of memories from transient acts vs. memories from systematic, periodic repetitions of those acts.

Here we offer a unifying framework with a taxonomy of function and differentiable levels of intent, awareness, and control paired with a new statistical platform for personalized analyses of natural behaviors. This new model aims to capture and characterize the micro-fluctuations in the gross data of our biorhythms that traditional approaches throw away as noise through grand averaging and “*one size fits all*” methods. Our approach allows integration of multilayered hierarchical signals and provides the means to differentiate re-entrant contributions from multilayered exo- and endo-afference. This can help our self-realization of embodied agency as the spontaneous transformation of mental intent into physical volition. We invite the reader to consider this new model for embodied cognition and offer novel avenues to bridge the currently disconnected fields of motor control and cognitive phenomena.

## Figures and Tables

**Figure 1 jpm-10-00076-f001:**
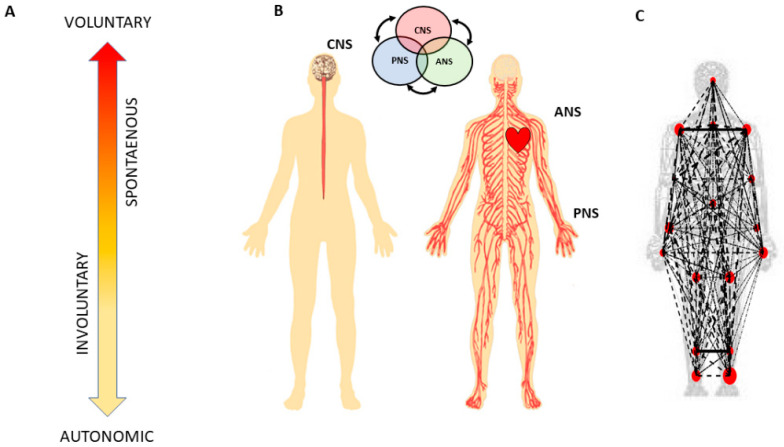
Defining quantitative aspects of agency for the study of embodied cognition. (**A**) A highly simplified schematics reflecting the phylogenetically orderly taxonomy of nervous system functions involving different levels of voluntary control (intent) ranging from deliberate to spontaneous movement segments, to involuntary motions and autonomic control. Levels correspond to three fundamental muscle types (skeletal for voluntary, smooth for involuntary, and cardiac for autonomic.) Multi-layered signals contributing from each of these layers are proposed to differentially contribute to the sense of action ownership and to the overall sense of agency via sensory consequences preceded by different levels of intent. (**B**) Contributions of the central and peripheral nervous systems (CNS and PNS, respectively), including the autonomic nervous system (ANS), can be tracked in a closed loop that helps the autonomous realization of intended thoughts into physical actions under volitional control. (**C**) Network connectivity analyses of kinematics and heart biorhythmic signals encompassing these levels of control enable the study of agency through objective quantitative methods.

**Figure 2 jpm-10-00076-f002:**
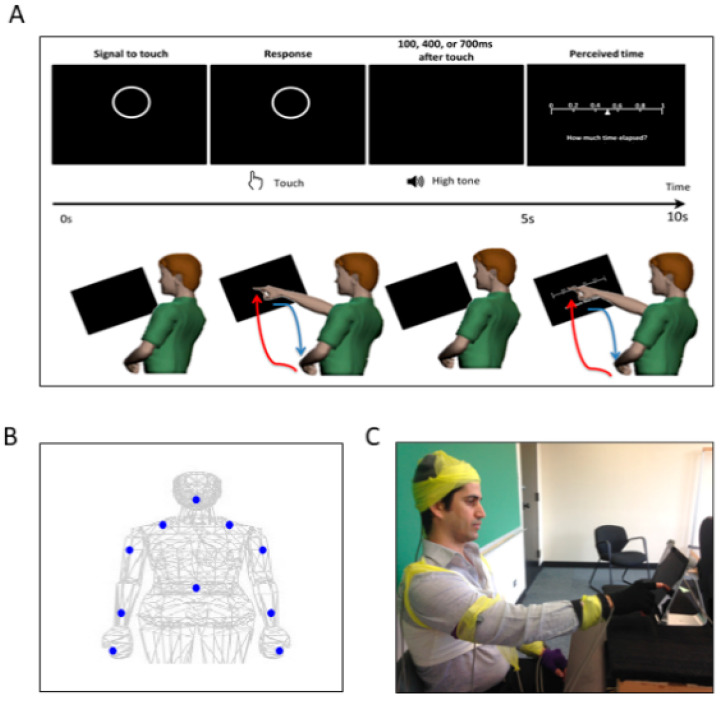
Experimental assay and instrumentation setup. (**A**) Experimental procedure. In a single trial, the participant was presented with a display screen, as shown on the top panel. During the first 5 s, the participant was presented with a circle as a prompt to touch the circle on the screen. After the touch, the participant heard a tone. The duration between the touch and the tone was randomly set to be 100 ms, 400 ms, or 700 ms. In the next 5 s, the participant was presented with a sliding scale, where s/he indicated how long the time was perceived to have elapsed between the touch and the tone, by touching the corresponding number on the scale. For each trial, the participant made two pointing gestures—one to touch the circle and another to indicate their time estimation on the sliding scale. Such pointing gesture was composed of a forward reaching segment (red) and a backward retracting segment (blue), as shown in the bottom panel. (**B**) Motion capture sensor positions. The sensors were attached on the following body parts: center of the forehead, thoracic vertebrate T7, right and left scapula, right and left upper arm, right and left forearm, non-performing hand, and the performing hand’s index finger. (**C**) Snapshot of the experiment. During the experiment, the participant was seated in front of the tablet screen to perform the tasks, and wired sensors were secured with athletic tape.

**Figure 3 jpm-10-00076-f003:**
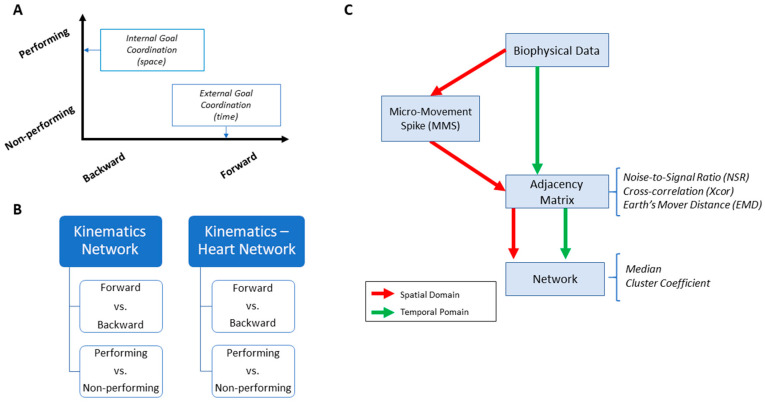
Overview of analytics pipeline. (**A**) Behavioral assay to quantify ranges of motor intent along two axes to highlight externally and internally defined goals. Along the former, motions are classified across time based on the end-effector’s movement, ranging from backward–spontaneous (lower motor intent) to forward–deliberate (higher motor intent) motions. Along the other axis, motions are classified across locations of the body, based on the proximity to the end-effector, from non-performing side of the body parts (lower motor intent) to the performing side including the end-effector (higher motor intent). Note, the two axes are not necessarily orthogonal as the schematics imply. (**B**) Two types of network analyses were made. Within the kinematics network, kinematics data served to compare patterns of variability from movement segments of higher level of intent (deliberately aimed at the goal) and movement segments with lower level of intent (spontaneous retractions of the hand to rest, without instructions), including as well comparison of patterns from the performing and non-performing parts of the body. Within the kinematics-heart network, a similar comparison was made, with a layer of autonomic function added, using signals from the EKG sensors. (**C**) For the spatial domain of connectivity analysis, raw biophysical data (biorhythms) co-registered from multiple layers of the peripheral and autonomic nervous systems were converted to MMS and used to compute pairwise similarity/synchronicity metrics to build adjacency matrices to represent weighted/undirected graphs. For the temporal domain, the raw biophysical data were directly used to build adjacency matrices. For both domains, with the obtained adjacency matrices, network connectivity analyses combined with non-linear dynamical systems approaches were used to identify self-emerging kinematic synergies and various indexes to enable objective quantification of the embodied cognition phenomena.

**Figure 4 jpm-10-00076-f004:**
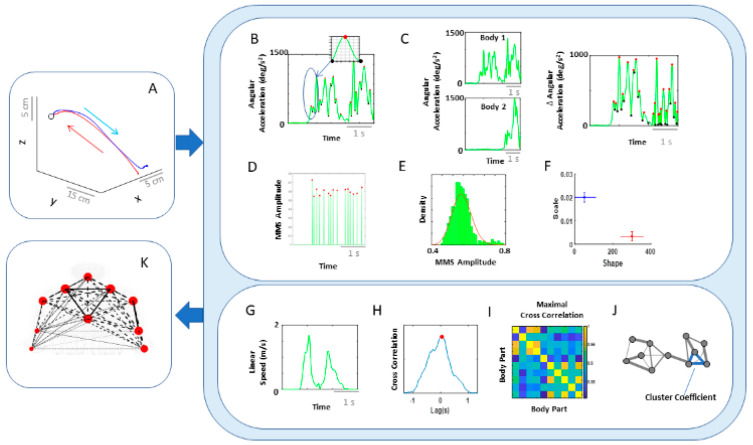
Analytical pipeline and visualization methods for the kinematics network. (**A**) Representative movement trajectory of the performing hand during a pointing motion to a target (denoted by a small open circle). Each trial was comprised of a forward–deliberate (red) and backward–spontaneous (blue) segment. These could be automatically separated by the speed and distance criteria (see Figure A2). (**B**) Time series of angular acceleration of the performing hand’s index finger during a typical pointing task. To examine kinematics-based connectivity, we used the angular acceleration time series, focusing on the moment by moment fluctuations in waveform amplitude. Here, peaks (maxima) and valleys (minima) are shown in red and black dots, respectively. The inset shows a zoomed-in picture of a single angular acceleration segment (i.e., two local minima and a single peak in between, used for standardization described in Equation (1). (**C**) Pairwise absolute difference in waveform was obtained and standardized using Equation (1). The resulting waveform provided the input to obtain MMS. (**D**) MMS train scaling the waveform amplitude for a typical pointing task. All standardized spike amplitude values from (**B**) and (**C**) were maintained, while all non-spike values were set to 0. (**E**) Frequency histogram of MMS amplitudes fitted to a Gamma probability distribution function (PDF) using maximum likelihood estimation (MLE). (**F**) The empirically estimated Gamma parameters (shape and scale) were obtained and plotted on a Gamma parameter plane, with marker lines representing the 95% confidence interval. Noise-to-signal ratio (NSR) (i.e., fitted Gamma scale parameter) were later used for comparison between motor segments and different performing side. (**G**) Representative time series of linear speed of the performing hand’s index finger in one trial. (**H**) Pairwise cross-correlation between two body parts. (**I**) Adjacency matrix obtained from all pairwise maximal cross-correlation across all body parts under consideration, to represent a weighted undirected graph. (**J**) Connectivity metrics (e.g., clustering coefficient) were used to quantify patterns of temporal dynamics. (**K**) Network connectivity analyses to unveil self-emerging clusters, where nodes correspond to each body part. For the spatial domain, NSR derived from MMS amplitudes of angular accelerations were visualized as node size, and NSR derived from MMS amplitudes of pairwise absolute difference in angular acceleration as edge thickness. For the temporal domain, cluster coefficients were visualized as node size, and median cross-correlations as edge thickness.

**Figure 5 jpm-10-00076-f005:**
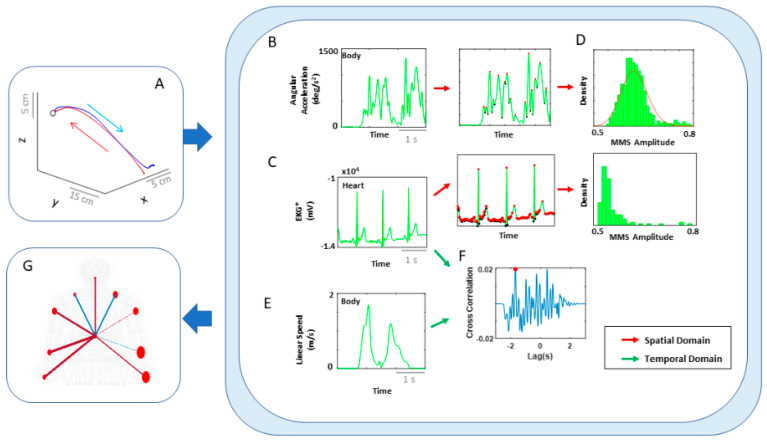
Analytical pipeline and visualization methods for the kinematics-heart network. (**A**) Typical movement trajectory of the performing hand position, while performing a single pointing action towards a target. Each trajectory was separated into forward–deliberate (red) and backward–spontaneous segments (blue) according to hand–target updated distance and near-zero-speed value (see Figure A2 for details). (**B**) Angular acceleration time series of the hand during a typical pointing task. MMS amplitudes from the angular acceleration time series were extracted for each body part. (**C**) Filtered EKG time series during a pointing task. MMS amplitudes from the filtered EKG time series were extracted. (**D**) Histograms of compiled MMS amplitudes. For spatial analysis, pairwise EMD was computed between histograms from each body part and heart activity. (**E**) Linear speed time series of the performing hand. For temporal analysis, linear speed kinematics time series was used. (**F**) Cross-correlation between a single body part’s linear speed and filtered EKG signal. For each trial, cross-correlation was computed between a pair of filtered EKG and a single body part’s linear speed time series, and the maximal value (red dot) and its corresponding lag values were extracted. (**G**) Visualization of connectivity. Network connectivity was visualized, where node size represented the EMD between the corresponding pair of body part and heart signals (i.e., spatial metric), and edge thickness represented the median cross-correlation values between the signal pairs (i.e., temporal metric). The edge colors were visualized, such that red would indicate EKG signals temporally leading linear speed signals, and blue would indicate linear speed leading EKG signals.

**Figure 6 jpm-10-00076-f006:**
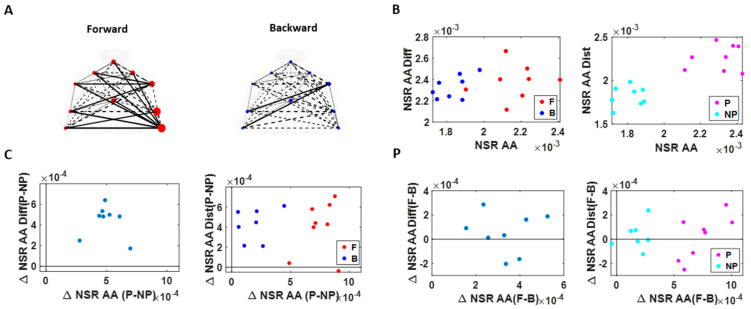
NSR signatures during pointing can differentiate the levels of intent. Comparison includes forward–deliberate vs. backward–spontaneous segments and performing vs. non-performing effector. (**A**) Network visualization of a right-handed representative participant. Node size is represented by the NSR derived from the corresponding body part’s kinematics time series, and edge thickness is represented by the NSR of the absolute difference in kinematics between the corresponding pairs of body parts. Node size and edge thickness are graphed in the same scale across different movement segments (i.e., forward and backward segments). (**B**) NSR for different movement segment and sides. Each dot is the median NSR values for each participant’s different movement segments (left) and different sides (right) from the unitless MMS derived from the angular acceleration (AA) fluctuations in amplitude. The x-axis denotes the NSR from individual body part’s kinematics (NSR AA), and y-axis denotes the NSR from the MMS derived from the absolute pairwise body parts’ difference (NSR AA Diff). Generally, for the former (NSR AA) measure, NSR is higher during a forward segment (F; red) than during a backward segment (B; blue), and on the performing side (P; pink) than on the non-performing side (NP; cyan). (**C**) NSR difference between performing vs. non-performing side. Left panel shows the NSR median difference between the performing and non-performing side for each participant, denoted as a single marker. Right panel shows the NSR median difference between the performing and non-performing side for the forward motion (F; red) and backward motion (B; blue). When the difference between the performing and non-performing side is examined separately for each motion segment, the NSR AA difference is wider during forward motion segments (F; red) than during backward motion segments (B; blue). (**D**) NSR difference between forward vs. backward movement segment. Left panel shows the NSR median difference between the forward and backward motion segments for each participant, denoted as a single marker. Color scheme as in (**B**).

**Figure 7 jpm-10-00076-f007:**
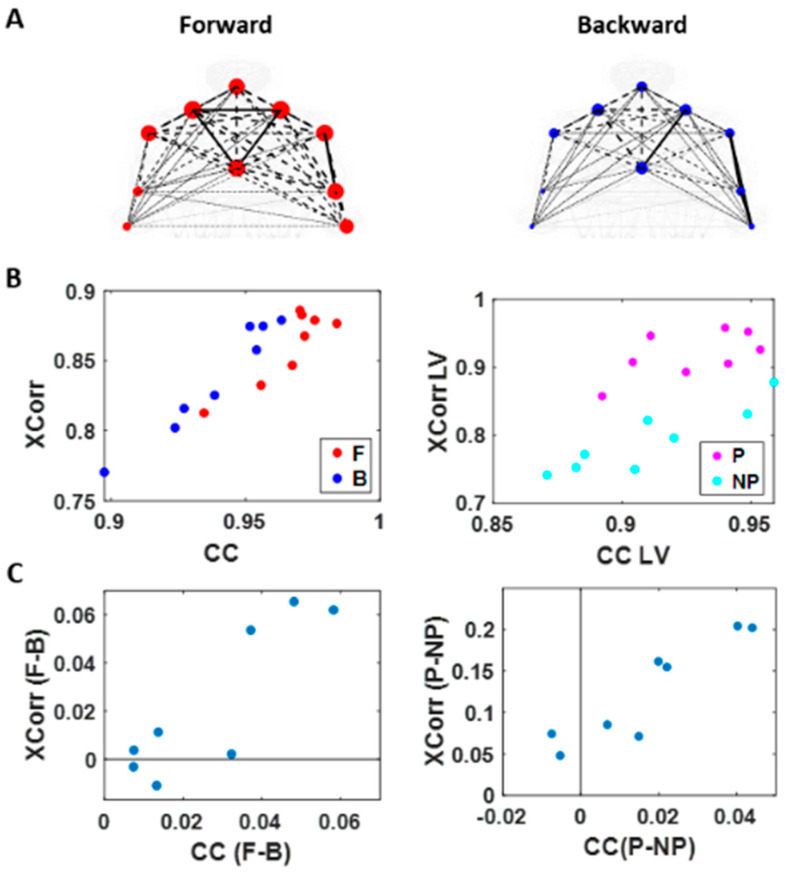
Network connectivity metric (cluster coefficient) and median cross-correlation differentiates between levels of intent. (**A**) Network visualization of a representative right-handed participant. Cross-correlation is represented by the line weight and cluster coefficient (CC) by the node size, during the forward (left) and backward movement segments (right). (**B**) Median cross-correlation (y-axis; Xcorr) and CC (x-axis) of linear speed for each participant’s movement segment (left) and different sides (right). Forward motions (red) and performing side (pink) exhibits higher cross-correlation and CC values than backward segments (blue) and non-performing side (cyan). (**C**) Median cross-correlation and CC difference for different movement segments (left) and different sides (right). Each participant’s data is denoted as a single marker. Higher motor intent tends to show higher cross-correlation and CC values.

**Figure 8 jpm-10-00076-f008:**
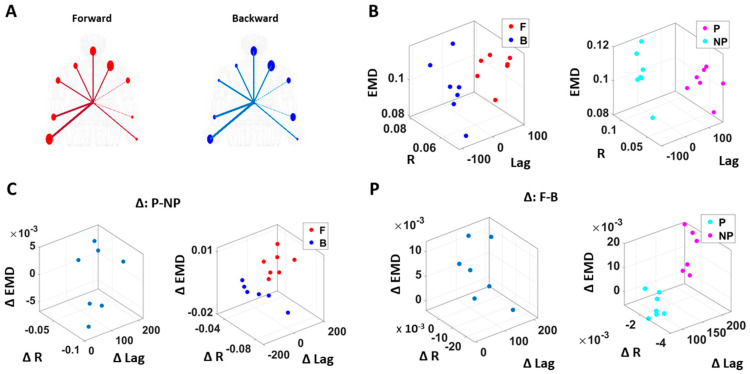
Differentiation of spatial and temporal connectivity within the kinematics-heart network according to levels of motor intent. (**A**) Network visualization of a right-handed representative participant. 1/EMD is represented by the node size, and median correlation is represented by the line weight. The color of edges indicates the temporal directionality between signals, where red indicates that heart leads the body linear speed and blue indicates that body linear speed leads the heart signals. (**B**) Median EMD (z-axis) and correlation (x-axis) and lag (y-axis) for each participant’s movement segment (left) and different sides (right). There is an overall pattern where higher motor intent (denoted by red for forward motions, and pink for performing side) is exhibited by lower correlations and EKG leading the kinematics signal (i.e., lag is positive value). (**C**) Median EMD, correlation, and lag difference for different sides (left), and this difference separated by movement segment (right). We find a pattern where pairwise EMD show higher differentiation under higher motor intent on the performing side, but only when forward motion was made. (**D**) Median EMD, correlation, and lag difference for different movement segments (left), and this difference separated by sides (right). We find a pattern where pairwise EMD show higher differentiation under lower motor intent on the non-performing side, but only when the backward motion was made.

**Table 1 jpm-10-00076-t001:** Summary of the connectivity results, where symbols ^1^ are shown to indicate which category shows higher values.

		**Kinematics (AA) Network**			
		**Forward**	**Backward**	**Performing**	**Non-Performing**
Spatial	NSR AA	o		o	
	NSR AA Diff			o	
Temporal	Cross-Correlation	Δ		o	
	Cluster Coefficient	o		Δ	
		**Kinematics (LS)-Heart Network**			
		**Forward**	**Backward**	**Performing**	**Non-Performing**
Spatial	EMD	Δ (P) ^2^	-	Δ (F) ^3^	o (B) ^4^
Temporal	Cross-Correlation		Δ		o
	Lead *^,5^	EKG	LS	EKG	LS

^1^ o indicates that it is higher for every participant; Δ indicates that it is higher for most participants. ^2^ Forward–deliberate motions have higher EMD only on the performing (P) side. ^3^ Performing side has higher EMD only during forward–deliberate (F) motions. ^4^ Non-performing side has higher EMD only during backward–spontaneous (B) motions. ^5^ Lead* shows which signal leads between the 2 signals.

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
