# Peer review of "The Autonomic Nervous System Differentiates between Levels of Motor Intent and End Effector"

_jpm, 2020, doi:10.3390/jpm10030076_

Round 1
Reviewer 1 Report
Authors did an excellent job by presenting a paper that attempts to bring together the field of embodied cognition with the physiological parameters of the nervous system. Overall, all sections are well written and described, the methodological part is very clear, in terms of procedure and statistical analysis, although some additions are needed to improve the introduction and theoretical discussion:
speaking of the link between property and agency, you should take into consideration previous works that have described very well the link between the motor system and the sense of agency, both in natural and experimental / guided actions (see della Gatta et al 2016 and 2017). Furthermore, to increase the quality of insights at the clinical level, the authors should also integrate the work of Garbarini et al. 2015 (focused on the relationship between property and agency in neurological disease) and Riva et al. 2017 (focused on the concept of incorporated medicine and artificial intelligence models for the new treatment in some psychiatric disorders).
“Decreased motor cortex excitability mirrors own hand disembodiment during the rubber hand illusion” (della Gatta F., Garbarini F., Puglisi G., Leonetti A., Berti A., Borroni P.). eLife, 2016 Sept; 5:e14972. DOI:
10.7554/eLife.14972
“Drawn Together: When Motor Representations Ground Joint Actions” (della Gatta F., Garbarini F., Rabuffetti M., Viganò L., Butterfill S., Sinigaglia C.). Cognition, 2017 Apr. https://doi.org/10.1016/j.cognition.2017.04.008
“When Your Arm Becomes Mine: Pathological Embodiment Of Alien Limbs
Using Tools Modulates Own Body Representation” (Garbarini F., Fossataro C., Berti A., Gindri P., Romano D., Pia L., della Gatta F., Maravita A., Neppi-Modona M.). Neuropsychologia, 2015 Apr; 70:402-13.
doi:10.1016/j.neuropsychologia.2014.11.008
"Riva G., Serino S., Di Lernia D., Pavone E. F., Dakanalis A. (2017), Embodied Medicine: Mens Sana in Corpore Virtuale Sano. Front. Hum. Neurosci. 11:120. doi: 10.3389/fnhum.2017.00120"
Author Response
Authors did an excellent job by presenting a paper that attempts to bring together the field of embodied cognition with the physiological parameters of the nervous system. Overall, all sections are well written and described, the methodological part is very clear, in terms of procedure and statistical analysis, although some additions are needed to improve the introduction and theoretical discussion:
speaking of the link between property and agency, you should take into consideration previous works that have described very well the link between the motor system and the sense of agency, both in natural and experimental / guided actions (see della Gatta et al 2016 and 2017).
We thank the reviewer for the comments. We have carefully read this literature and added the relevant references in the Introduction on page 2 line 45. We also point the reviewer to the place where we added them using a comment box
Furthermore, to increase the quality of insights at the clinical level, the authors should also integrate the work of Garbarini et al. 2015 (focused on the relationship between property and agency in neurological disease) and Riva et al. 2017 (focused on the concept of incorporated medicine and artificial intelligence models for the new treatment in some psychiatric disorders).
We have added these to the Discussion on page 19 line 761
Additional references incorporated in the paper:
“Decreased motor cortex excitability mirrors own hand disembodiment during the rubber hand illusion” (della Gatta F., Garbarini F., Puglisi G., Leonetti A., Berti A., Borroni P.). eLife, 2016 Sept; 5:e14972. DOI:
10.7554/eLife.14972
“When Your Arm Becomes Mine: Pathological Embodiment Of Alien Limbs
Using Tools Modulates Own Body Representation” (Garbarini F., Fossataro C., Berti A., Gindri P., Romano D., Pia L., della Gatta F., Maravita A., Neppi-Modona M.). Neuropsychologia, 2015 Apr; 70:402-13. doi:10.1016/j.neuropsychologia.2014.11.008
"Riva G., Serino S., Di Lernia D., Pavone E. F., Dakanalis A. (2017), Embodied Medicine: Mens Sana in Corpore Virtuale Sano. Front. Hum. Neurosci. 11:120. doi: 10.3389/fnhum.2017.00120"
We added this one too since seems relevant
Burin, D., et al., Are movements necessary for the sense of body ownership? Evidence from the rubber hand illusion in pure hemiplegic patients. PLoS One, 2015. 10(3): p. e0117155.
We did not add the Della Gatta et al. (2018) Drawn together: when motor representations ground joint actions because although it is a very interesting work, we did not study joint action here. However, we have an upcoming paper where this work will be very relevant to cite, so we thank the reviewer for pointing us to this literature.

Reviewer 2 Report
If I have understood correctly, through skin sensors and the performance of an electrocardiogram, the authors obtain a reflection of the activity of the CNS, PNS and ANS in the face of intentional and unintentional movements in left and right-handed men and women. In the abstract the authors conclude: "We find that when the action is intended, the heart signal leads the body kinematics signals; but when the action segment spontaneously occurs without instructions, the heart signal lags the bodily kinematics signals"
The approach of this type of research is very interesting. The work shows a complex scenario. They study the dynamics between voluntary intentional movements and involuntary unintended movements involving the ANS. When the action is intentional, the cardiac signal leads body dynamics, but when the action is automatic, the cardiac signal delay body dynamics.
The results reflect the complexity of the organism's response, using signals from the CNS, PNS and ANS. The integration of these signals to give rise to a motor response will depend, according to the authors, on whether the movement is intentional or unintended.
I have some minor considerations:
There is no mention in the abstract of what the title suggests.
Due to the number and type of participants (2 males and 7 females) (two left-handed and 7 right-handed), it seems difficult to reach conclusions about differences between sexes and differences between manual laterality.
The whole article is very interesting but the reader unfamiliar with the subject, attracted by the title, would appreciate in the discussion a more understandable section that would clarify the expectations created after reading the title "The Autonomic Nervous System Differentiates Between Levels of Motor Intent and Hand Dominance".
What aspects of the results support that title? What are the implications of those results?
Where is the discussion of the results that differentiate between left and right handed?
Are there differences between the result of connecting CNS, PNS and ANS signals between intentional and unintended movements when comparing left-handed to right-handed?
Are there differences between males and females?
Is there nothing in the literature that connects ANS to lateralized manual ability?
Author Response
If I have understood correctly, through skin sensors and the performance of an electrocardiogram, the authors obtain a reflection of the activity of the CNS, PNS and ANS in the face of intentional and unintentional movements in left and right-handed men and women. In the abstract the authors conclude: "We find that when the action is intended, the heart signal leads the body kinematics signals; but when the action segment spontaneously occurs without instructions, the heart signal lags the bodily kinematics signals"
The approach of this type of research is very interesting. The work shows a complex scenario. They study the dynamics between voluntary intentional movements and involuntary unintended movements involving the ANS. When the action is intentional, the cardiac signal leads body dynamics, but when the action is automatic, the cardiac signal delay body dynamics.
The results reflect the complexity of the organism's response, using signals from the CNS, PNS and ANS. The integration of these signals to give rise to a motor response will depend, according to the authors, on whether the movement is intentional or unintended.
I have some minor considerations:
There is no mention in the abstract of what the title suggests.
We thank the reviewer for the comments provided. We have modified the abstract to better reflect the title (on line 22 we point the reviewer to the change using a comment box)
Due to the number and type of participants (2 males and 7 females) (two left-handed and 7 right-handed), it seems difficult to reach conclusions about differences between sexes and differences between manual laterality.
We note that we made the mistake of using the word “dominant” about the performing limb. We meant the limb-hand performing the task in relation to the opposite limb. We have corrected this in the title and throughout the paper to avoid this confusion. Indeed, we did not study laterality or hand dominance here, but rather end effector performance. We have now changed the title to The Autonomic Nervous System Differentiates Between Levels of Motor Intent and End Effector. We have corrected all figures too and throughout the MS we have been careful to replace dominant with performing hand.
The whole article is very interesting but the reader unfamiliar with the subject, attracted by the title, would appreciate in the discussion a more understandable section that would clarify the expectations created after reading the title "The Autonomic Nervous System Differentiates Between Levels of Motor Intent and Hand Dominance".
What aspects of the results support that title?
The focus on the differentiation of levels of intent by the autonomic signals appear on line 653 or page 17 in the Discussion. We point the reviewer to the spot in the Discussion where we do so, using a comment box
What are the implications of those results?
Our interpretation of the findings starts at line 662 and the implications on line 677-719
Where is the discussion of the results that differentiate between left and right handed?
We have added a subsection to that end, starting at line 720 of page 18
Are there differences between the result of connecting CNS, PNS and ANS signals between intentional and unintended movements when comparing left-handed to right-handed?
There are differences when comparing performing vs non-performing end effector. We quantified them in the NSR and in the network connectivity patterns and describe the results in sections 3.1 (starting on line 490) and 3.2 starting on line 537.
Are there differences between males and females?
We did not examine here differences between males and females, though it is an interesting question for future research
Is there nothing in the literature that connects ANS to lateralized manual ability?
Not to the best our knowledge, but we did not study laterality. We studied performing vs non-performing end effectors. The word dominant was erroneously used in the prior version. We have corrected this in this version.
We thank the reviewer once more for the useful feedback and for taking the time to help us make our work clearer to the readership of the journal.

Reviewer 3 Report
The manuscript entitled “The autonomic nervous system differentiates between levels of motor intent and hand dominance” examined the ability of autonomic signals (i.e., heart rate, micro-movements) to differentiate intended from automatic processes. Movements were examined using a sophisticated wireless motion capture system and the study utilized three within conditions varying in cognitive load (control, low, and high cognitive load). The movement analysis procedures were very detailed, though issues with lag reduced accuracy. Although the number of participants analyzed (n = 7) was very small, this issue seemed to be lessened with a substantial number of trials combined for each participant during analyses. The use of arcane language and misapplication of the term “spontaneous” substantially weakens the impact of the study. In spite of these substantial problems, I believe that this manuscript has promise, and that the methodological and analytical approach is novel and could add to the literature if revisions are made.
I have a love-hate attitude with the arcane vocabulary used in the Introduction of this manuscript. I admire the use of concise vocabulary, but the writing goes too far. The commonality of arcane terms (e.g., nascent) and more complex concepts (e.g., geometric invariants) creates a situation that is punishing or unapproachable to many readers. When a common term or phrase is equally detailed and concise to an arcane one, the more common term should be used. Thus, the authors need to reconsider the use of arcane vocabulary in contrast to a writing style that is more approachable to students and the general scientific community. Otherwise, the manuscript will have a low impact simply because readers will be turned-away by the difficulty of reading it. I suggest revising the Introduction to using vocabulary that is more approachable to students and those new to the field. After all, it is students who will lead the way in using newer technologies like those described in this study, and you do not want them to be turned off by an Introduction that is overly difficult to comprehend.
The authors defined “spontaneous movements” as return and change movements that follow touching the circle target (line 169). This definition is very problematic because it goes against the basic definition of spontaneous. Spontaneous is impulsive and without premeditation or an external stimulus. The return/change movement is not impulsive, but a return response to touching the circle. The return/change movement is also premeditated because it functions to return the hand to a waiting position for the next task. I am not aware of prior research using “spontaneous” to describe a return movement that occurs immediately following a touch task. Thus, the term “spontaneous” needs to be removed and instead "automatic" or “return/backward movement” terms need to be used. I would be fine with “spontaneous” being used when referring to the nondominant hand movements as long as the hand was never involved in the task. Given the actual dominant hand return movements examined, however, “spontaneous” is not an accurate term to characterize the return movements analyzed.
In lines 56 – 60, it is stated that unintended motions produce different signatures of variations via spontaneous feedback. Of course all movements involve spontaneous feedback, but it is not clear how the signatures are different. It should be clarified how this “signature” differs between intended and more automatic return movements.
I have several concerns over Figure 1. Figure 1A is clearly relevant to the study, but is overly simplistic to illustrate all levels of neuromotor control. Figure 1A is confusing since it is not clear how involuntary differs from autonomic. Also the figure does not clearly show a progression from deliberate to spontaneous (I do not equate “deliberate” with “voluntary”). Last, it would be more informative if this figure showed how the combined contributions of parallel CNS, PNS, and ANS systems had different strengths to differentiate spontaneous and involuntary actions. One approach would be to have 3 vertical arrows (CNS, PNS, ANS) that changed color to illustrate how an action moved from voluntary to involuntary. Figures 1B & 1C are not as necessary to the study. Thus, I suggest removing Figure 1B and 1C, and expanding 1A.
An important aspect of the study design was cognitive load, yet the authors did very little to analyze nor discuss cognitive load. The Discussion needs to explain the impact of cognitive load, and how it related to the results. At minimum, limitations should discuss why cognitive load was not examined and analyzed in more detail.
Author Response
The manuscript entitled “The autonomic nervous system differentiates between levels of motor intent and hand dominance” examined the ability of autonomic signals (i.e., heart rate, micro-movements) to differentiate intended from automatic processes. Movements were examined using a sophisticated wireless motion capture system and the study utilized three within conditions varying in cognitive load (control, low, and high cognitive load). The movement analysis procedures were very detailed, though issues with lag reduced accuracy. Although the number of participants analyzed (n = 7) was very small, this issue seemed to be lessened with a substantial number of trials combined for each participant during analyses. The use of arcane language and misapplication of the term “spontaneous” substantially weakens the impact of the study. In spite of these substantial problems, I believe that this manuscript has promise, and that the methodological and analytical approach is novel and could add to the literature if revisions are made.
We thank the reviewer for the comments and try to address the issues the reviewer raises below.
I have a love-hate attitude with the arcane vocabulary used in the Introduction of this manuscript. I admire the use of concise vocabulary, but the writing goes too far. The commonality of arcane terms (e.g., nascent) and more complex concepts (e.g., geometric invariants) creates a situation that is punishing or unapproachable to many readers. When a common term or phrase is equally detailed and concise to an arcane one, the more common term should be used. Thus, the authors need to reconsider the use of arcane vocabulary in contrast to a writing style that is more approachable to students and the general scientific community. Otherwise, the manuscript will have a low impact simply because readers will be turned-away by the difficulty of reading it. I suggest revising the Introduction to using vocabulary that is more approachable to students and those new to the field. After all, it is students who will lead the way in using newer technologies like those described in this study, and you do not want them to be turned off by an Introduction that is overly difficult to comprehend.
We have revised the Introduction to reflect the reviewer’s concerns. We mark the paragraphs where we have changed the term geometric invariants and unfolded what they are.
The authors defined “spontaneous movements” as return and change movements that follow touching the circle target (line 169). This definition is very problematic because it goes against the basic definition of spontaneous. Spontaneous is impulsive and without premeditation or an external stimulus.
We have clarified the use of the term spontaneous in the introduction. In the context of full loop pointing motions, we introduced the paradigm as far back as 2010 and characterized the two phases of the reach in 2011, using complex sports routines in boxing and in the tennis serve. “Spontaneous” movements of the overt kind refer to uninstructed segments that coexist with deliberate segments (staged to a goal). The term is different from “automatic” movements in that automatic movements may also include segments that are intended to a goal and segments that are unintended, spontaneously performed. Up to our work, the pointing motions were studied with a sole focus on the forward segment to the target. We introduced the characterization of spontaneous retractions in the precise sense that they are uninstructed and do not pursue a goal. They are consequential to the goal directed segment ending at the target. The fundamental difference between the two segments is that the forward one is invariant to speed changes and to changes in other dynamics of the action (e.g offsetting the mass distribution of the arm or of the body’s center of mass). They are also invariant at the joint angle level featuring postural paths that are less perturbed by dynamic changes than the retracting ones. We have also compared the retracting ones when they are instructed vs when they are uninstructed, and they have different stochastic signatures. As such, we make this distinction based on an objective physical quantification of the trajectories of the hand and of the full arm in the postural domain. Here we also go beyond the performing limb and explore the spontaneous retractions through the rest of the upper body. We point the reviewer to line 53 of the introduction, where we further explain the difference bet covert spontaneous motions and overt ones.
The return/change movement is not impulsive, but a return response to touching the circle. The return/change movement is also premeditated because it functions to return the hand to a waiting position for the next task.
There is a fundamental difference between instructed and uninstructed return motions. Here we did not instruct the return and in this sense, it is spontaneously self-initiated. Furthermore, the motion away from the target does not have to be towards the initial position where the hand initiated the reach from. It is free to be towards any location. The point is that such location is not a well-defined target. The segment is merely a connecting motion between a goal directed one (intentional and deliberately performed to an instructed target) and the such next motion that will repeat in the experiment. To disambiguate the term spontaneous, we now use the term uninstructed-spontaneous vs instructed-deliberate.
I am not aware of prior research using “spontaneous” to describe a return movement that occurs immediately following a touch task.
We introduced the term spontaneous and characterized the motions at more than one level (end effector and posture space, following a model published in the J of Neurophys in 2020 by Torres and Zipser)
Other related work:
Two classes of movements in motor control (in Experimental Brain Research)
https://pubmed.ncbi.nlm.nih.gov/22038712/
Impaired Endogenously Evoked Automated Reaching in Parkinson's Disease (in the J of Neuroscience)
https://www.jneurosci.org/content/31/49/17848.short
Applications to sports
Signatures of Movement Variability Anticipate Hand Speed According to Levels of Intent (in Behavioral and Brain Functions)
https://behavioralandbrainfunctions.biomedcentral.com/articles/10.1186/1744-9081-9-10
Applications to Autism diagnoses and treatments
Give spontaneity and self-discovery a chance in ASD: Spontaneous peripheral limb variability as a proxy to evoke centrally driven intentional acts (in Frontiers in Integrative Neuroscience)
https://www.frontiersin.org/articles/10.3389/fnint.2013.00046/full
Autism: the micro-movement perspective (also in Front Int Neuroscience)
https://www.frontiersin.org/articles/10.3389/fnint.2013.00032/full
Strategies to Develop Putative Biomarkers to Characterize the Female Phenotype with Autism Spectrum Disorders (in J of Neurophys)
https://journals.physiology.org/doi/pdf/10.1152/jn.00059.2013
Applications to Parkinson’s Disease
Spatial Orientation-Priming Impedes Rather than Facilitates the Spontaneous Control of Hand-Retraction Speeds in Patients with Parkinson's Disease (in PLoS One)
https://journals.plos.org/plosone/article?id=10.1371/journal.pone.0066757
Applications to Schizophrenia
Schizophrenia: The Micro-movements perspective (in Neuropsychologia)
https://www.sciencedirect.com/science/article/pii/S0028393216000270
Thus, the term “spontaneous” needs to be removed and instead "automatic" or “return/backward movement” terms need to be used. I would be fine with “spontaneous” being used when referring to the nondominant hand movements as long as the hand was never involved in the task. Given the actual dominant hand return movements examined, however, “spontaneous” is not an accurate term to characterize the return movements analyzed.
Please see above
In lines 56 – 60, it is stated that unintended motions produce different signatures of variations via spontaneous feedback. Of course all movements involve spontaneous feedback, but it is not clear how the signatures are different. It should be clarified how this “signature” differs between intended and more automatic return movements.
Not all feedback is spontaneous. That is precisely the point of the experimental assay and results. Figure 1A shows different levels of motor control that have been well characterized by our lab. There are 50 peer reviewed publications covering the topic in detail here (spanning 20 years of rigorous mathematical and computational work) https://sensorymotorintegrationlab.com/publications
I have several concerns over Figure 1. Figure 1A is clearly relevant to the study, but is overly simplistic to illustrate all levels of neuromotor control. Figure 1A is confusing since it is not clear how involuntary differs from autonomic.
It is indeed a simplified version of all levels of neuromotor control, yet these are levels that have been well characterized as we now afford non-invasive means to do so. The caption explicitly says this is an oversimplified schematic of neuromotor control levels but they are motivated by the three different fundamental muscle types that our lab studies in analyzes of the human genome https://www.biorxiv.org/content/10.1101/2020.07.19.210971v1.
Involuntary movements are different from autonomic in that although both are involuntary in nature, one is random (e.g. ticks) and the other is pacemaker-like with highly periodic rhythms. Furthermore, they have fundamentally different origins as one is primarily linked to organs lined by smooth muscles and / or skeletal muscles, and the other to cardiac muscles.
Examples of a characterization of a type of involuntary motions uses involuntary head motions during the resting state of fMRI experiments in various studies of autism in relation to neurotypical controls
Aging with Autism Departs Greatly from Typical Aging (in the journal of Sensors from MDPI)
https://www.mdpi.com/1424-8220/20/2/572
Motor noise is rich signal in autism research and pharmacological treatments (in the open access nature journal of scientific reports)
https://www.nature.com/articles/srep37422
Among others…
Also the figure does not clearly show a progression from deliberate to spontaneous (I do not equate “deliberate” with “voluntary”).
As we mention in the caption this is a simplified schematic of a more comprehensive taxonomy based on levels of control that we have quantified and characterized using thousands of participants over the years.
Last, it would be more informative if this figure showed how the combined contributions of parallel CNS, PNS, and ANS systems had different strengths to differentiate spontaneous and involuntary actions. One approach would be to have 3 vertical arrows (CNS, PNS, ANS) that changed color to illustrate how an action moved from voluntary to involuntary. Figures 1B & 1C are not as necessary to the study. Thus, I suggest removing Figure 1B and 1C, and expanding 1A.
We appreciate and respect the reviewer’s opinion and recommendations. However, we are going to respectfully leave the figure as is. This is a proposition to the field rather than a statement written on stone. We are working hard to characterize these levels using noninvasive means.
An important aspect of the study design was cognitive load, yet the authors did very little to analyze nor discuss cognitive load. The Discussion needs to explain the impact of cognitive load, and how it related to the results. At minimum, limitations should discuss why cognitive load was not examined and analyzed in more detail.
We have published a full paper focusing on the issue of cognitive loads and now mention this in the Discussion, while addressing the reviewer’s comment.
Characterization of Sensory-Motor Behavior Under Cognitive Load Using a New Statistical Platform for Studies of Embodied Cognition (in Frontiers in Human Neuroscience)
https://www.frontiersin.org/articles/10.3389/fnhum.2018.00116/full
